

# Estimating the likelihood of roadway pluvial flood based on crowdsourced traffic data and depression-based DEM analysis

Arefeh Safaei-Moghadam[1], David Tarboton[2], and Barbara Minsker[1]

[1]Department of Civil and Environmental Engineering, Southern Methodist University, Dallas, TX, USA
[2]Department of Civil and Environmental Engineering, Utah Water Research Laboratory, Utah State University, Logan, Utah, USA

**Correspondence:** Arefeh Safaei-Moghadam (asafaeimoghadam@smu.edu)

**Abstract.** Water ponding and pluvial flash flooding (PFF) on roadways can pose a significant risk to drivers. Furthermore, climate change, growing urbanization, increasing imperviousness, and aging stormwater infrastructure have increased the frequency of these events. Using physics-based models to predict pluvial flooding at the road segment scale requires notable terrain simplifications and detailed information that is often not available at fine scales (e.g., blockage of stormwater inlets). This brings uncertainty into the results, especially in highly urbanized areas where micro-topographic features typically govern the actual flow dynamics. This study evaluates the potential for flood observations collected from Waze–a community-based navigation app–to estimate the likelihood of PFF at the road segment scale. We investigated the correlation of the Waze flood reports with well-known flood observations and maps, including the National Flood Hazard Layer (NFHL), high watermarks, and low water crossings data inventories. In addition, highly-localized surface depressions and their catchments are derived from a 1-meter-resolution bare-earth digital elevation model (BE-DEM) to investigate the spatial association of Waze flood reports. This analysis showed that the highest correlation of Waze flood reports exists with local surface depressions rather than river flooding, indicating that they are potentially useful indicators of PFF. Accordingly, two data-driven models, Empirical Bayes (EB) and Random Forest (RF) regression, were developed to predict the frequency of flooding, a proxy for flood susceptibility, for three classes of historical storm events (light, moderate, and severe) in every road segment with surface depressions. Applying the models to Waze Data from 150 storms in the City of Dallas showed that depression catchment drainage area and imperviousness are the most important predictive features. The EB model performed with reasonable precision in estimating the number of PFF events out of 92 light, 41 moderate, and 17 severe storms with 0.84, 0.85 and 1.09 mean absolute errors, respectively. This study shows that Waze data provides useful information for highly localized PFF prediction. The superior performance of EB compared to the RF model shows that the historical observations included in the EB approach are important for more accurate PFF prediction.

## 1 Introduction

This study developed and tested a new data-driven framework for short-term flash flood likelihood estimation at the scale of road surface depressions based on crowdsourced traffic data. Flash flooding is considered one of the most hazardous natural disasters that affect people worldwide (Kousky, 2018). Analysis of flash floods over the contiguous United States shows that



flash flood frequency and property damage have increased in the past two decades (Ahmadalipour and Moradkhani, 2019). Pluvial flash flooding (PFF) is defined as localized floods caused by an overwhelmed natural or engineered drainage system (Carter et al., 2015; Rosenzweig et al., 2018). PFF can reduce the reliability of roadway networks by decreasing capacity, increasing travel time, reducing safe speed, and increasing accident risks and deaths through lane submersion (Agarwal et al., 2005; Suarez et al., 2005; Smith et al., 2004).

Most urban flood studies have focused on fluvial and coastal flooding rather than PFF. Rosenzweig et al. (2018) identified three reasons for pluvial flooding being less studied: 1—It is assumed that stormwater infrastructure, such as sewers, culverts, and pumps, are sufficient to prevent pluvial flooding, 2—Pluvial flooding is believed to be a nuisance with minimal impacts, and 3—Lack of monitoring data to capture short-duration precipitation over small urban watersheds.

In the past, stormwater minor system (curbs, gutters, inlets, pipes, and channels) have been designed to minimize nuisance
hazards associated with a 10-year or less recurrence interval rainfall (U.S. Department of Transportation FHWA, 1979). More recent roadway facilities are designed and evaluated for 50-year and 100-year events (Mark and Marek, 2011), but in older urban areas, undersized conveyance systems remain (Jack et al., 2021). With climate change, growing urbanization, and increasing imperviousness, the frequencies of extreme rainfall events and nuisance flooding are increasing (United Nations., 2019); Hemmati et al., 2021, 2020), leading to increased risks from pluvial flooding. Mobility disruption is a noticeable conse-
quence of PFF (Douglas et al., 2010; Yin et al., 2016; Coles et al., 2016; Li et al., 2018). For example, Pregnolato et al. (2017) estimated that a driver facing 10 cm of standing water must not drive faster than 40 km/hr to maintain safe driving, stopping, and steering without loss of control.

In order to warn drivers about rapidly changing flash flood conditions, high-resolution predictive models are needed at navigational scale (road segment and intersection). Simplified terrain models, such as rapid flood spreading model (RFSM)
(Lhomme et al., 2008), height above nearest drainage model (HAND) (Nobre et al., 2011), and hierarchical filling and spilling models (Zhang and Pan, 2014; Chu et al., 2013; Wu et al., 2019; Samela et al., 2020) can estimate inundation extent in less complex terrains where the dynamics of flow, velocity, and momentum are negligible (Teng et al., 2017). Statistical methods are also able to predict flooding by analyzing historical observations, however, since they learn from the past, updating procedures are required to make them adaptive to accelerated future changes as they are built upon the assumption that similar conditions
in the future will cause flooding. A notable advantage of statistical PFF models is their ability to capture impacts of unobserved variables and uncertainties from historical observations, as well as the ability to rapidly update the models as new data become available and system dynamics change. Haghighatafshar et al. (2020) suggested that designing stormwater infrastructure based on storm recurrence intervals is ambiguous while statistical models can provide the basis of a more resilient system by taking uncertainties of vulnerability and hazard of pluvial flooding into account. Many studies have investigated statistical flood
modeling to predict flooding by applying statistical and machine learning methods such as classification models, Bayesian frameworks, and Random Forest models (Tien Bui and Hoang, 2017; Solomatine and Ostfeld, 2008; Tehrany et al., 2013; Zahura et al., 2020). Other studies have combined deterministic physics-based models with statistical models for forecasting applications (Li and Willems, 2020; Zhao et al., 2018).



Empirical and data-driven models require flooding observation data with high spatio-temporal resolution. The average duration of flash flooding events in the United States has been 3.5 hours during the last two decades (Ahmadalipour and Moradkhani, 2019), limiting the applicability of aerial imagery to obtain sufficiently frequent flash flooding observations. To fill this data gap, there is increasing interest in the application of newer "crowdsourced" data into flood modeling, monitoring, and impact assessment (Molinari et al., 2018; Gaitan et al., 2016; See, 2019; Assumpcao et al., 2018; Praharaj et al., 2021; Helmrich et al., 2021; Zhu et al., 2022; Liu et al., 2021; Schnebele et al., 2014). Previous crowdsourced flood data studies have involved engaging citizens in collecting four types of data: streamflow or rain gauge readings, videos, text messages, and image postings (Li and Willems, 2020; Assumpcao et al., 2018; Zhu et al., 2022; Liu et al., 2021; Schnebele et al., 2014; Le Coz et al., 2016; Smith et al., 2017; Cervone et al., 2015; Wang et al., 2018; Pereira et al., 2020; Moy De Vitry et al., 2019). Also,Zhu et al. (2022) and Liu et al. (2021) applied artificial intelligence techniques to extract flooding waterlogging from microblog information shared in crowdsourcing apps. A big challenge in using crowdsourced data is identifying the accurate location and flood extent from posted pictures, videos, and texts. However, even with the challenges mentioned above, researchers have concluded that integrating crowdsourced data into flood models improves the overall performance and timeliness of forecasts, hence increasing flood hazard awareness (Assumpcao et al., 2018; Goodrich et al., 2020).

The majority of studies have implemented crowdsourced data into physics-based models as complementary data for model setup, calibration, validation, and data assimilation (Zahura et al., 2020; Assumpcao et al., 2018; Smith et al., 2017). However, physics-based models can be limited in flood prediction at road segment scales due to highly complex and interconnected variables that contribute to flooding in urban environments (Coles et al., 2016; Rafieeinasab et al., 2015). Micro topographic features, steep slopes, and varying surface materials can generate different types of flow regimes at small spatial scales. Dual-drainage hydrodynamic models that couple equations for the underground sewer system and surface flow, require detailed layouts of urban drainage systems that can be of varying quality, particularly in older urban areas where PFF is most prevalent (Haghighatafshar et al., 2020; Smith et al., 2017; Sadler et al., 2018; Berndtsson et al., 2019). Finally, catchments that drain into roadways are often very small and ungauged, leading to further uncertainties in estimating road inundation (Versini et al., 2010). Hence accurate high-resolution real-time physics-based hydrodynamic modeling in urban areas is computationally extensive and rarely considered feasible (Mignot et al., 2006; Sanders et al., 2020).

In this study, we address these gaps and limitations of PFF probability estimation on roadways by incorporating crowdsourced navigation data from the Waze navigation app as highly localized flood observations into high-resolution data-driven models that can be updated and implemented rapidly to provide near-real-time navigational warnings. The framework developed has three steps. In the first step, road surface depressions and their upstream catchments are delineated from a high resolution digital elevation model using simplified flow-routing and hierarchical fill spill approaches. In the second step, two statistical and machine learning models—Empirical Bayes (EB) and random forest (RF)— are developed and tested to predict PFF frequency using roadway, catchment, depression, and rainfall characteristics. In the third step, probability of roadway flooding and flood maps are generated that could be disseminated to navigation software. To our knowledge, this study is the first to develop real-time PFF likelihood maps at road segment scales using data-driven models and crowdsourced traffic



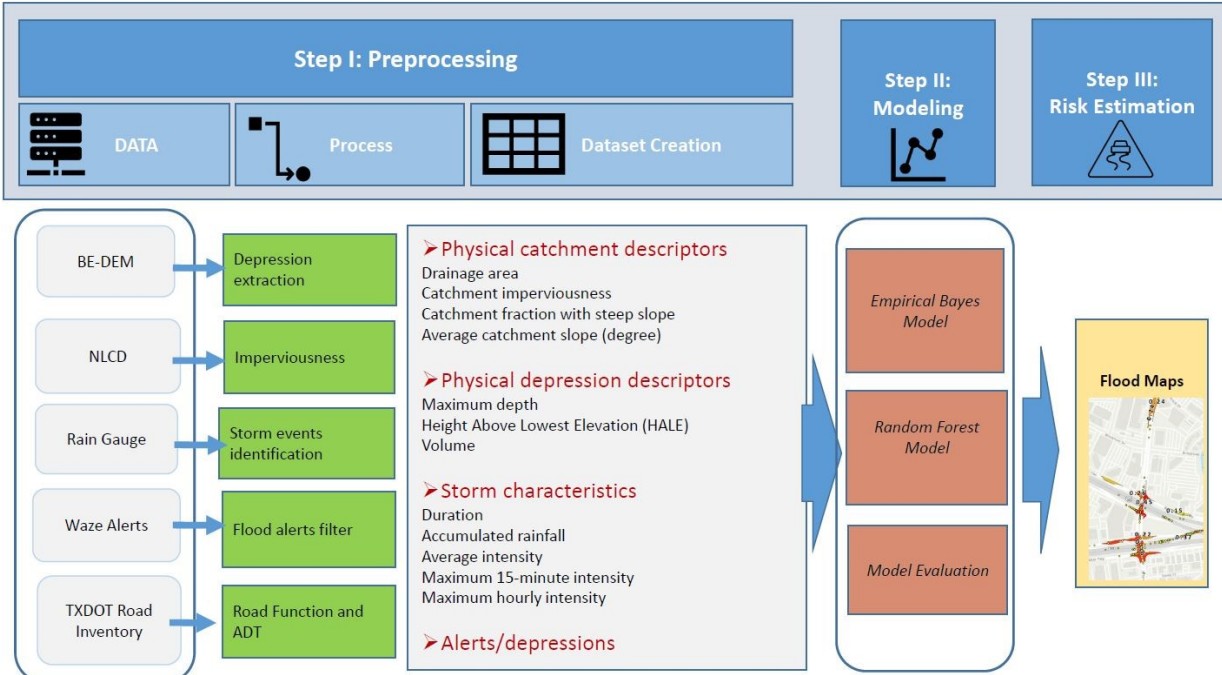

**Figure 1.** Methodology framework (basemap from ESRI-2021)

data. With the widespread use of smartphones and crowdsourced applications, this study shows the benefits of integrating crowdsourced data and statistical modeling approaches into roadway flood awareness and management systems.

## 2 Methodology

95

The three steps of the framework developed are shown in Figure 1. The first step involves data preprocessing to create the dataset needed for modeling. The second step fits statistical and machine learning models to the historical dataset, and the third step performs the roadway flooding likelihood estimation for future storms. These steps are described in more detail in sections below.

### 2.1 Step I: Preprocessing

100

The dataset preprocessing in Step I includes three primary components that are described in detail in the sub-sections below and depicted in Figure 1. First, road surface depressions and their upstream catchments are delineated. Second, storm events and their characteristics are determined from continuous rain gauge observations; third and last, flood alerts are assigned to corresponding depressions and storm events.





### 2.1.1 Depression extraction

The first step of data preprocessing is to find road surface depressions that are prone to PFF. Generally, surface depressions are defined as the difference between the hydrologically-conditioned digital elevation model (DEM) (Lindsay and Dhun, 2014) and the raw DEM. In hydrologically-connected DEM elevations of internally draining sinks are raised to form a flat area that can drain to downstream. Locating surface depressions in a highly urbanized terrain is challenging due to micro-topographic and underground features (such as curbs, stormwater inlets, etc.) that determine the actual flow path. In addition, using a high-resolution DEM (1-meter) introduces hierarchical depressions with different orders of magnitude in spatial scale, from highly localized (minor pits) to surface depressions that cover more than one neighborhood (residual depressions). Therefore, a nested hierarchy of depressions must be considered to extract depressions compatible with urban features.

In this paper, the "sink evaluation" tool of the ArcHydro toolbox (Djokic et al., 2011) is utilized to extract a nested hierarchy of surface depressions. The sink evaluation tool scans the bare earth DEM (BE-DEM) and characterizes low-lying cells. The process of local depression extraction is an iterative process that examines each sink, raises the elevation of low-lying cells to fill the sink, and then reapplies the process on the resulting DEM. This procedure is depicted in Figure 2. In the first sink evaluation step, Level-1 depressions are delineated and raised (Figure 2-a). In the second step, the DEM resulting from the first level fill (Figure 2-e, red areas) is evaluated and Level-2 depressions are delineated. This process can be repeated until the area is fully hydrologically-conditioned and no higher-level depressions remain. The number of steps required in this process is dependent on the resolution of the DEM and the complexity of the depressions in the landscape.

Due to the complexity of urban terrain, the spatial scale of depressions at each hierarchy level is quite variable and depressions at the same level can be as large as a neighborhood or as small as a pothole. Therefore, we did not set an automated stopping criterion in terms of depression level or depth for the depression filling process. Instead, depression at all hierarchical levels were extracted and those depressions that best align with urban features manually selected. Figure 2-e shows 10 depressions (L1-1 to L1-7, L2 1, L2-2, and L3-1) extracted on a road segment with three depression levels. Level-1 depressions and L2-2 appear as small potholes or single cell pits that could be DEM errors, but regardless are too small to cause traffic disruptions. However, L2-1 aligns with road curbs and gutters and could cause traffic disruptions. Therefore, L2-1 is manually selected as the smallest depression that is prone to PFF and could affect traffic flow on this road segment. (Note that L3-1 includes L2-1, hence it will be filled only after L2-1 has filled and disrupted traffic flow already. Hence, L3-1 does not need to be included in the model for traffic navigation purposes.)

### 2.1.2 Physical depression and catchment descriptors

After delineating road surface depressions, physical descriptors of depressions and their upstream catchments are computed as follows. Two sets of characteristics, summarized in Table 1, are defined for every depression that is selected in the previous extraction step: physical depression descriptors (PDD) and physical catchment descriptors (PCD)(Kalantari et al., 2014). PDD features describe the depression topography that is likely to affect water accumulation. These features are area, average depth assuming the depression is filled, and the height of road DEM cell elevations above the lowest elevation of the depression





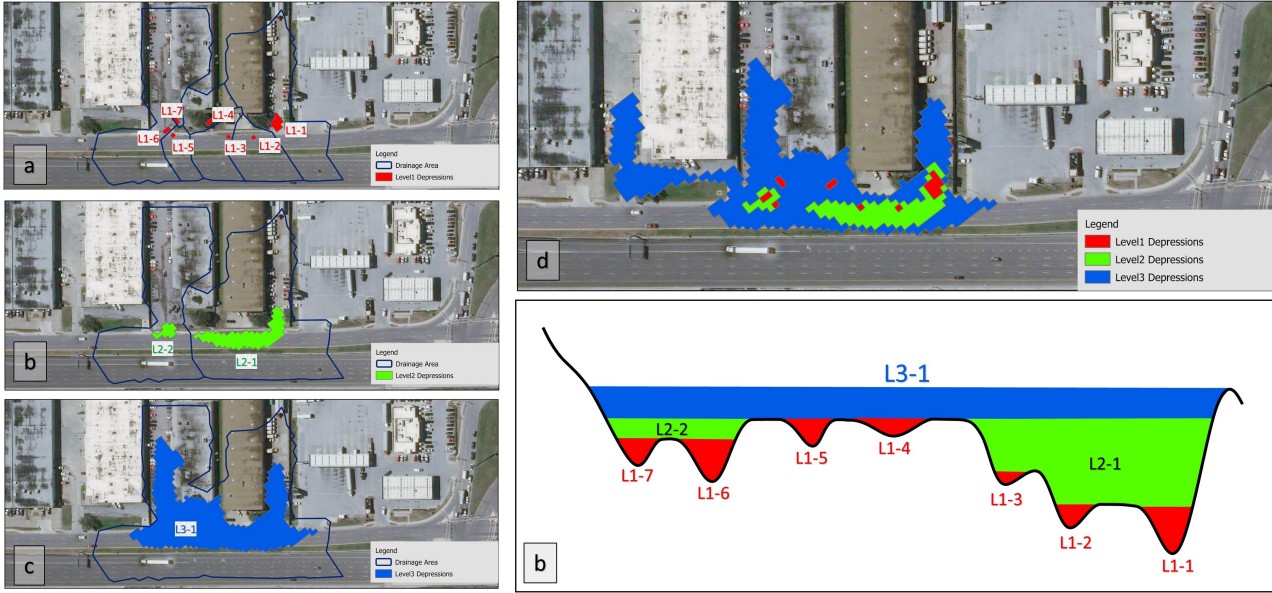

**Figure 2.** Hierarchical filling of surface depressions (basemap from ESRI-2021)

(hereafter called Height Above Lowest Elevation, or HALE). The HALE feature indicates which terrain cells would be inundated first and what depth is required for accumulated water to reach the road surface. Figure 3 shows a schematic of the HALE and depth features. The PCD features are derived from the upstream catchment that drains into each depression. The extracted features are average slope, fractions of the upstream catchment with a steep slope (defined as steeper than 8%), percentage of imperviousness, and the net drainage area, which is computed using Equation 1:

$$NetDA = Log(CA) \times I \tag{1}$$

Where:

$NetDA$ is the net drainage area in $log(m^2)$,

$CA$ is the catchment area in $m^2$, and

$I$ is the percentage imperviousness of the catchment based on the National Land Cover Dataset (NLCD)

$Log(CA)$ was used in this equation reflecting that the rate of changes in runoff volume for large catchments is expected to be less than for small catchments. This can happen since the larger the drainage area is, the higher are the impacts of infiltration, loss and stormwater drainage that we are not considering in this analysis.




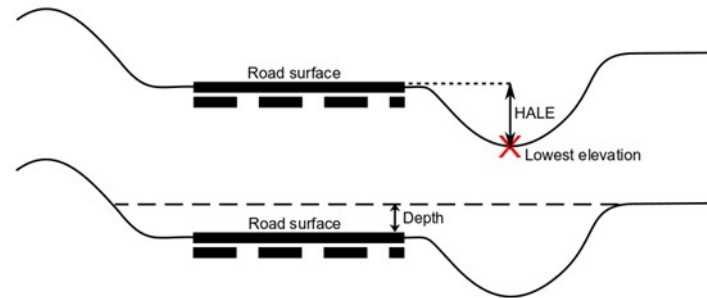

**Figure 3.** Schematic of HALE and depth features

### 2.1.3 Traffic exposure

Crowdsourced data are generated by volunteer contributions, which results in more data availability on roads with higher traffic volumes. Therefore, including a feature in the model that captures roadway traffic exposure to flooded areas is necessary to consider the likelihood of reporting a flooded depression. For this purpose, two additional variables are included in the framework (Table 1): (1) the natural logarithm of Annual Daily Traffic (ADT) and (2) the road function as defined by the Texas Department of Transportation (TX-DOT).

**Table 1.** Physical depression/catchment descriptors

|  | Depression descriptor | Definition | Unit | Source |
|---|---|---|---|---|
| PDD | Depression area | The area of the road surface that the depression covers | Square meters | DEM processing |
|  | Average depth | The average depth assuming that the depression is filled | Meters | DEM processing |
|  | Maximum depth | The maximum depth assuming that the depression is filled | Meter | DEM processing |
|  | Depression volume | The volume that fills the depression | Cubic meters | DEM processing |
|  | Minimum volume | The volume that generated 6-in depth on the road | Cubic meters | DEM processing |
|  | HALE | The average height of the road above the lowest elevation of the depression | Meters | DEM processing |
| PCD | Net drainage area | Proxy to the runoff generated from the upstream catchment | Square meters | DEM processing |
|  | Upstream imperviousness | Average imperviousness fraction of the upstream catchment | Percentage | NLCD |
|  | Upstream steep slope | The fraction of the catchment area that has a slope steeper than 8 percent | Percentage | DEM processing |
|  | Average upstream slope | The average slope of upstream catchment | Degree | DEM processing |
| Road | Log ADT | Natural logarithm of the ADT | Vehicles/day | TX-DOT Inventory |
|  | Road function | The function of the road as 1: interstate, 2: Freeway and Expressway, 3: Principal Arterial, 4: Minor Arterial, 5: Major Collector, 6: Minor Collector, 7: Local | N/A | TX-DOT Inventory |



### 2.1.4  Storm event definition and storm clustering

Raw precipitation data are obtained from Automated Surface Observing Systems (ASOS) stations in continuous 5-minute
interval rain pulse observations. To predict the probability of depression flooding during a storm of particular severity, independent storm events must be derived from the continuous data. In this study, the Minimum Inter-event Time (MIT) method is used to define independent storm events. The MIT approach defines a storm event as rainfall that follows and is followed by a minimum dry (rainless) period called the Minimum Inter-event Time. The MIT value can be calculated using different approaches. A reasonable estimate of the MIT value is the lag-time at which the serial autocorrelation between rain pulses
reaches a pre-set low threshold and remains steady(Asquith et al., 2005). In this study, the MIT value is diagnosed using the correlogram method to visualize the autocorrelation of a rain pulse timeseries to find the lag time that makes a rain pulse independent of its preceding rain pulses. After defining independent storm events, storm characteristics, including accumulated precipitation, duration, average intensity, and maximum 15-minute, 30-minute, and hourly intensities, are calculated.

Depending on storm events' severity and terrain characteristics, storms can produce similar patterns of depression PFF. To
capture this phenomenon, storms are clustered into classes using the storm characteristics. For storm clustering, agglomerative hierarchical clustering is applied using a bottom-up approach that forms a single cluster for each storm event and successively merges clusters with the smallest distances between features. The benefit of using agglomerative clustering is that this algorithm is less sensitive to outliers (Edelbrock, 1979).

### 2.1.5  Waze data preprocessing

Waze is a GPS-based traffic navigation app that collects crowdsourced information about road conditions. The Waze app aggregates traffic incidents reported by its users as traffic alerts. Traffic alerts are geotagged points with two attributes that specify their lifetime: 'publish date' and 'last seen'. The Waze app has no pre-qualification for users to post a report, consequently not all of the flood-labeled alerts are reliable to be used as flood observations. Praharaj et al. (2021) showed that 71% of Waze flood alerts are reliable in Norfolk, Virginia. To investigate Waze alerts' authenticity, we matched flood-related alerts to the
most recent rainfall event and computed the delay between alerts' publishing and rainfall end-time. A temporal threshold can be found by analyzing the cumulative distribution of delays that determines whether a flood report is related to a storm event.

In addition to alert timing, we also compared the locations of Waze alerts to publicly available datasets of high-flood-risk locations, including the National Flood Hazard Layer (NFHL), high watermarks and low water crossings data inventories from the North Central Texas Council of Government (NCTCOG), and the road surface depressions computed as described in the
methodology section. The NFHL is a spatial dataset that uses river flood hazard information provided by the Federal Emergency Management Agency (FEMA) to generate flood hazard maps showing areas at high risk of flooding. We investigated the proximity of Waze alerts to the high flood risk locations to find the spatial accordance of flood alerts to these locations.




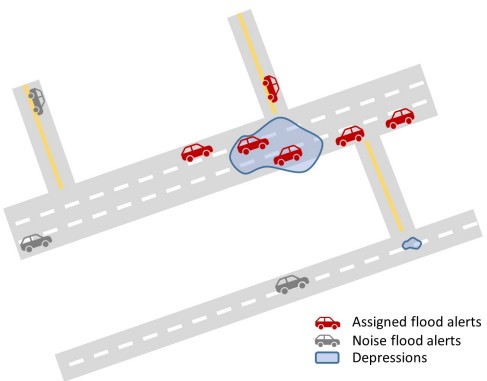

Assigned flood alerts
Noise flood alerts
Depressions

**Figure 4.** Alert assignment

One challenge in adopting Waze flood-related alerts as roadway PFF observations is assigning the alerts to the appropriate flooded location because the coordinates of alert points do not perfectly align with flooded location coordinates. The distance

between the flooded location and alerts depends on many unknown factors such as drivers' reaction times, direction, and sight distance; besides, posting a flood alert requires Waze users to complete three steps (three selections) in the app while driving or riding. Hence assigning flood alerts to the proper depression must be done carefully.

In this study, several independent individuals were asked to visually assess a map of historical flood alerts laid over surface depressions and assign alerts to depressions using the following criteria: a cluster of more than two flood alerts should be avail-

able near the depression and the depression must be distinct from other nearby surface depressions. Flood alerts posted from bridges and elevated highways are excluded since BE-DEM does not represent bridge surfaces. Figure 4 shows a schematic example of alerts that can be assigned to the depicted depression and some that should remain unassigned because they are isolated and too far from a depression.

## 2.2 Step II: Modeling

Pluvial flooding on any given surface depression can be modeled as a Bernoulli trial of flood failure (i.e., non-flooded) or success (i.e., flooded). Assuming that the probability of being flooded is smaller than the non-flooded situation and that the likelihood of flooding in a particular storm event for each depression is independent of the probability of flooding for other depressions, a random variable $y_{i,j}$ will define the count of successes (flooding) out of the N trials (N storm events of cluster j) on depression $i$. The purpose of this study is to estimate the random variable $y_{i,j}$ using extracted topographic features, road

function, and storm severity. Both statistical and machine learning models are implemented to estimate $y_{i,j}$, namely Empirical Bayes and Random Forest. Table 2 summarizes the categories of pre-processed independent variables used in the modeling.




**Table 2.** Count dataset of PFF events

| | PDD & PCD | Road function (categorical) | Storm clusters | Count of Flooding |
|---|---|---|---|---|
| Depression $i^th$ | Topographic Features | Interstate<br>Freeway | Light | $y_{(i, j = light)}$ |
| | | Expressway<br>Principal Arterial | Moderate | $y_{(i, j = light)}$ |
| | | Minor Arterial<br>Major Collector | Severe | $y_{(i, j = light)}$ |

### 2.2.1 Empirical Bayes model

In a highly urbanized area there are numerous uncertain and unobserved site-specific features that affect localized PFF likelihood, such as storm inlet's age, capacity, and condition. For example, consider two road surface depressions (A and B) with

similar PDD, PCD, road type, and ADT that experience the same storm. Suppose Depression A is located in a neighborhood with lower infrastructure maintenance services, and its drainage system clogs more often. Then, despite similar descriptive features, higher flood frequency should be expected at depression A. The Empirical Bayes (EB) algorithm, a simplified and faster version of Bayes theory, takes advantage of the historical count of reported flood events from the Waze data to better reflect the impacts of these types of uncertain and unobserved variables. The EB approach has previously been implemented

in many fields to address the impacts of unobserved variables in estimating rare events, including hydrology. The EB method uses the joint global prior and site-specific counts and produces the posterior probability $y_i$ by employing a weighted average as shown in Equation 2 (Fill and Stedinger, 1998; Kuczera, 1982; Smith et al., 2014; Hauer et al., 2002; Lord et al., 2005; Strupczewski et al., 2001).

$$EB(y) = w \times \mu + (1 - w) \times y \qquad (2)$$

Where:

$w$ is the EB weight factor

$\mu$ is the expected flood frequency on depressions similar to a given depression, and

$y$ is the number of flood events on a given depression

The expected flood frequency for similar depressions ($\mu$) is the global prior probability distribution from a fitted regression

model, which in this study is a Negative Binomial regression model. The number of flood events ($y$) is the historical site-specific flood event observation from the Waze data.

### 2.2.2 Negative Binomial Distribution

Based on Waze flood observations, the variance of flood frequencies on depressions with similar PDD, PCD, road type, and ADT is assumed to be greater than the average of flood frequencies (i.e. $E(y) < Var(y)$). This assumption is appropriate given

the importance of unobserved variables on the PFF formation on roads such as storm inlet conditions. In other words, among




n similar surface depressions, k depressions, where k≪n experience flooding significantly more than average. This fact leads to an over-dispersed dataset where $E(y) < Var(y)$. Studies have shown that in the case of over-dispersed data, $y_i$ follows a Poisson distribution with the rate parameter $\lambda_i$, where $\lambda_i$ follows a Gamma distribution with the dispersion parameter $\phi$ and the rate parameter $\phi/\mu_i$. The resulting distribution is Poisson-gamma, also called the Negative Binomial (NB) distribution

67. The probability mass function of the NB distribution is given in Equations 3 and 4. Therefore, in this study, the expected flood frequency on similar depressions in the EB equation (Equation 2), is derived from a Negative Binomial (NB) regression model that is fit to the count dataset shown in Table 2. NB parameters ($\phi$ and $\beta_i$) are estimated using the Maximum Likelihood Estimation method.

$$P(y) = \frac{\Gamma(y+\phi)}{\Gamma(y+1)\Gamma(\phi)}(\frac{\phi}{\phi+\mu})^{\phi}(\frac{\mu}{\mu+\phi})^{y} \tag{3}$$

Where:

$\phi$ is the dispersion parameter of the NB distribution,

$y$ is number of flood events on depression $i$, and

$\mu$ is the expected flood frequency on a given depression based on similar depressions (Equation 4)

$$\mu = exp(\sum \beta_k x_k) \tag{4}$$

Where:

$\beta_k$ is the coefficient of $k^th$ regressor variable in fitted regression model

$x_k$ is the value of $k^th$ regressor on a given depression Model selection for the NB regression model is implemented using the Bayesian Information Criterion (BIC). In model selection, minimizing the BIC to the simplest model with the least number of exploratory variables is optimal. Reducing the BIC by adding more explanatory variables increases the risk of overfitting and

loss of generality. Equation 5 shows the calculation of BIC.

$$BIC = -2log(L) + KLn(n) \tag{5}$$

$L$ is the maximum likelihood of the model representing the overall fit of the model,

$K$ is the number of model parameters, and

$n$ is the sample size

It can be shown that the weight in the EB equation based on the NB regression is calculated as $\frac{\phi}{\mu+\phi}$, hence we can rewrite Equation 2 as Equation 6. For more information regarding the mathematics of deriving the weight factor of EB, refer to Zou et al. (2017).

$$EB(y) = \frac{\phi}{\mu+\phi}\mu + \frac{mu}{\mu+\phi}y \tag{6}$$

Where:

$\phi$ is the dispersion parameter of NB distribution,

$y$ is number of flood events on a given depression, and $\mu$ is the expected flood frequency on a given depression based on similar





depressions (Equation 4)

The EB model's predictive power is estimated using the mean absolute error (MAE). The MAE shows the average error of the fitted values across the observations. The lower the MAE, the better the EB estimates fit the observations. The MAE is calculated using Equation 7:

$$MAE = \frac{1}{n} \sum_{i=1}^{n} |y_i - \hat{y}_i| \tag{7}$$

Where:

$n$ is the sample size

$y_i$ is number of flood events on depression i, and

$\hat{y}_i$ is the EB predicted number of flood events on depression $i$

### 2.2.3 Random Forest

Random Forest (RF) is a supervised ensemble machine learning algorithm that uses multiple decision tree learners to increase predictive performance (Pedregosa et al., 2011). The final prediction of RF is the average prediction of all decision trees; each tree is built from a bootstrap sample of observations and a subset of features. The RF has been widely used for data-driven modeling in the field of water resources (Sadler et al., 2018). This algorithm can handle large and imbalanced datasets and is well known to be easy to train. An important strength of the RF is that its convergence rate is independent of noise and sparsity in the descriptive variables. RF models are useful for estimating the contribution of features in the target variable (in this case, flood frequency). The node impurity in each node of the RF is the measure of homogeneity of the target values at that node, which is the variance of target values in a regression problem. The normalized reduction in the node impurity achieved by adding a specific feature to a tree defines the importance of that feature. In RF, the average of importance of a feature in all trees weighted by the number of samples involved in each split is the overall feature importance.

In this study, RF regression is executed using the Scikit-Learn library in the Python environment (Pedregosa et al., 2011). The number of decision tree learners in the RF regression is optimized by the algorithm. For hyperparameter tuning and model selection, a randomized cross-validated grid search is applied on a wide range of model parameters and MAE is used to measure parameter performance and select the best-performing parameter set. The resulting parameters are then used to estimate the frequency of PFF at every depression for each storm class using Equation 8.

$$RF(y) = RF(PDD, PCD, road features, storm type) \tag{8}$$

Where:

$RF(y)$ is the random forest prediction of number of flood events on a given depression

### 2.2.4 Model Evaluation

To evaluate the performance of the proposed model, the following approaches are used. First, 80% of the historical data, randomly selected, are used in model training. Model testing is then implemented using the remaining 20% of the data held





out from the training process. The performance of the models is then assessed using the MAE of the predictions. In order to ensure that the models are stable and their performance does not change with different train-test sets, the models are trained

and evaluated for several randomly chosen training sets and the variation in their performance is considered in selecting the best models for the final step of the framework.

Then, to further assess the improvements in PFF event estimation using topographic and historical Waze observations, the EB and RF models are compared with three simple benchmark models. First, the average model (Equation 9) assumes that the average PFF counts from historical Waze observations apply to all depressions and all storms without considering storm type

and topographic feature. Second, the storm-based average model uses the average of the PFF count in each storm cluster without considering topographic features (Equation 10). Finally, a regression model is used that predicts PFF based on topographic, road type, and storm features but without implementing EB to update the prior probability (Equation 4).

$$p(i) = \sum_{i=1}^{n} \frac{y_i}{N_t} \tag{9}$$

Where:

$p(i)$ is the likelihood of flooding on depression $i$

$y_i$ is the number of reported floodings on depression $i$

$n$ is total number of depressions, and

$N_t$ is number of total storm events

$$p(i,j) = \sum_{i=1}^{n} \frac{y_{i,j}}{N_j} \tag{10}$$

Where:

$p(i,j)$ is the likelihood of flooding on depression $i$ and storm type $j$

$y_{(i,j)}$ is the number of reported floodings on depression i and storm type $j$, and

$N_j$ number of total storms of type $j$

### 2.3 Step III: Flood Probability Estimation

Finally, in Step III, the most accurate model from Step II is used to produce flood probability maps for every storm cluster across the region of interest. The probability of flooding is calculated using Equation 11.

$$p(i,j) = \frac{\hat{y}_{i,j}}{N_j} \tag{11}$$

Where:

$p(i,j)$ is the likelihood of flooding on depression i in a storm of type $j$

$y_{i,j}$ is the predicted number of floodings on depression $i$ and storm type of $j$, and

$N_j$ is the number of storms of cluster $j$



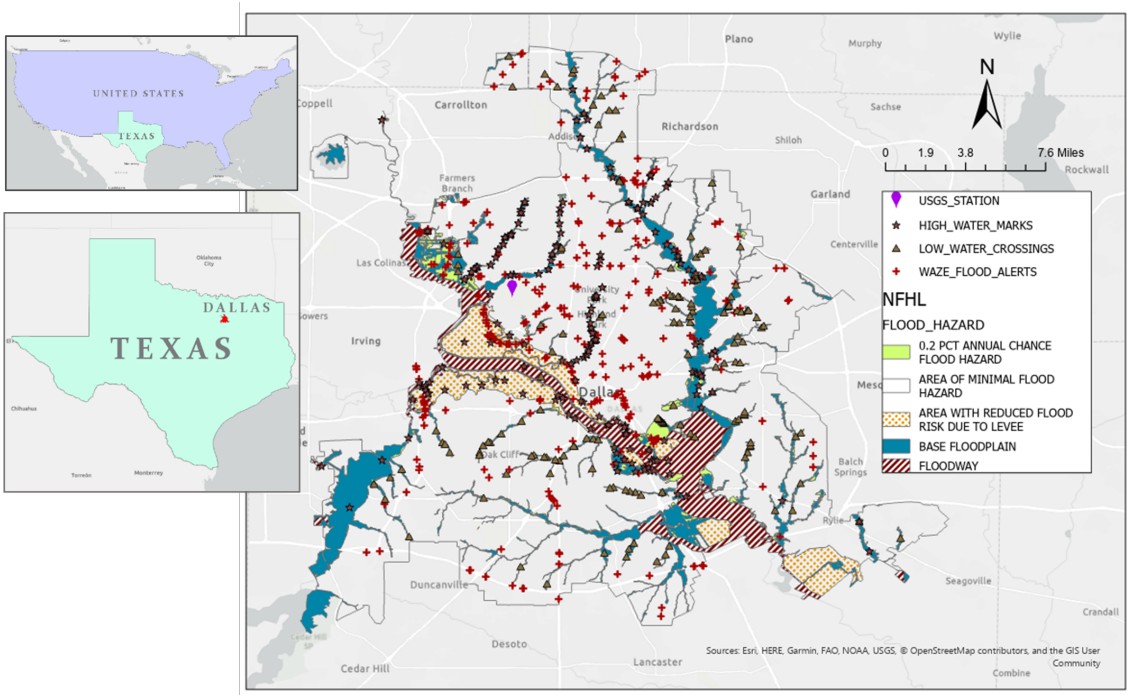

**Figure 5.** Study area and datasets (basemap from ESRI-2021)

## 3 Case study background and datasets

The described methodology was evaluated in the city of Dallas, Texas, USA (Figure 5), which is the third-largest city in Texas with a population of more than 1 million. Dallas elevation ranges from 137 to 168 meters (450 to 550 feet), and it is mostly flat.

According to the Texas Department of Transportation (TXDOT), almost 20 percent of crashes, equal to 248 vehicle crashes in the City of Dallas in 2018, happened on either standing water or wet road surface conditions. According to an analysis conducted by the First Street Foundation, flooding can expose 1841 miles of Dallas roadways (out of 6064 miles) to the risk of becoming impassable (F. S. Foundation, 2020)). However, currently available fire-rescue dispatch software, including that used by the Dallas Fire-Rescue Department (DFRD), assumes empty and dry roads for routing rescue vehicles. This has resulted in

rescue delays and occasional loss of life on flooded roadways, which provided the motivation for this study.



For this case study, several datasets were used. First, a 1-meter resolution Bare Earth Digital Elevation Model (BE-DEM) was obtained from the North Central Texas Council of Government (NCTCOG), which was derived from a Quality Level 2 Lidar survey performed by Digital Aerial Solutions, LLC, in 2018, under contract with the Unites States Geological Survey (USGS)/ National Resources Conservation Services (NRCS). The BE-DEM dataset's name is TX Pecos Dallas 2018 D19, with
horizontal accuracy of +/-0.682 meters at a %95 confidence level and non-vegetated vertical accuracy (NVA) of 0.196 meters.

For rainfall, 15-minute precipitation observations were obtained from the USGS ASOS station at Dallas Love Field Airport (DAL) (Figure 5). Precipitation observations from January 1st 2017 to March 1st 2020 were used. Next, the US Department of Agriculture's (USDA) National Land Cover Database (2016) (Homer and Fry, 2012) is used to extract catchment imperviousness. The imperviousness raster over Dallas has a 30-meter resolution and ranges from 0 to %100, with a mean of %33.87 and
standard deviation of %32.98.

Waze alerts were obtained from the NCTCOG, which is a Waze partner in the Waze Connected Citizen Program (CCP). The NCTCOG granted us access to the Waze data for the period of 2018-04-21 (the start of NCTCOG's Waze partnership) to 2020-03-20. Waze alerts are classified into seven main categories: accident, jam, construction, miscellaneous, hazard or weather (hazard-weather), road-closure, and others. The "hazard-weather" data itself is divided into several subcategories. Alerts in the
"flood" subcategory and ones which have any form of the word "flood" in their report description, such as "right lane flooded," are potentially flood-related and were included in this study, resulting in 5652 Waze alerts.

The locations of these Waze alerts were shown in Figure 5, along with the NFHL river flood zones. Figure 6a shows that the majority (around %70) of alerts during the study period were posted in areas with minimal river flood hazard, which comprise approximately %76 of the study area (Figure 6-b). Another %18 of the alerts were posted in areas of reduced river flood risk
due to levees, which were not breached during the study period. This indicates that PFF is likely the cause of most Waze alerts. To further investigate the potential causes of Waze flood alerts, the high-water marks inventory and low-water crossing dataset were obtained from the Texas Natural Resources Information System (TNRIS). The high-water marks inventory contains historic high water level reports from flooded water bodies or structures at 334 locations across the city of Dallas (Figure 5). The low-water crossing dataset includes 175 locations where surface water has crossed roads during high-flow conditions
(Figure 5). Analyzing Waze alert distances to the nearest high-water mark and low-water crossing shows that the vast majority of alerts are more than 200 meters from both low-water crossings and high-water marks (Figure 7). These findings show how complementary flood observations such as Waze data are needed to assess roadway conditions more comprehensively than available official datasets. Thus, in order to predict local roadway PFF, it is necessary to consider local surface depressions as low-lying areas where surface runoff can accumulate during storms.

**4   Case Study Data Pre-Processing**

**4.1   Depression Extraction**

Almost 380,000 surface depressions were extracted over the city of Dallas in three steps of hierarchical filling (described in the methodology). Only 315 depressions are located on roads and deeper than 6 inches. Among these 315 depressions, 191


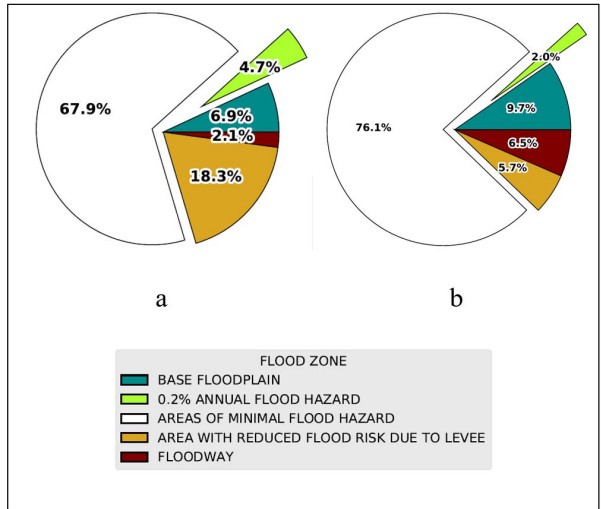

**Figure 6.** a: Flood alerts over NFHL flood zones b: Distribution of NFHL flood zone over the study area

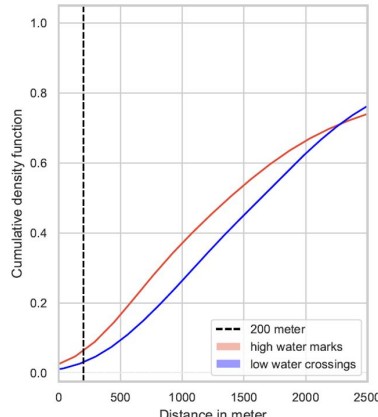

**Figure 7.** Cumulative density of alert distances to closest high-water mark and low-water crossing

depressions were proximal to reliable Waze flood alerts more than twice. To consider only chronically flooding areas, the rest
of this analysis is focused only on these 191 surface depressions.

## 4.2 Storm Event Definition

As described in the methodology section, the autocorrelation of rain pulses defines the optimal MIT for independent storm
events. Figure 8 shows that the autocorrelation coefficient first reaches a low value and remains steady at a lag time of 9 hours;
accordingly, MIT = 9 hours is chosen to convert the continuous precipitation data into independent storm events. Using MIT=9
hrs, 236 independent storm events are extracted from January 1st 2017 to March 1st 2020. For each storm event, duration; total

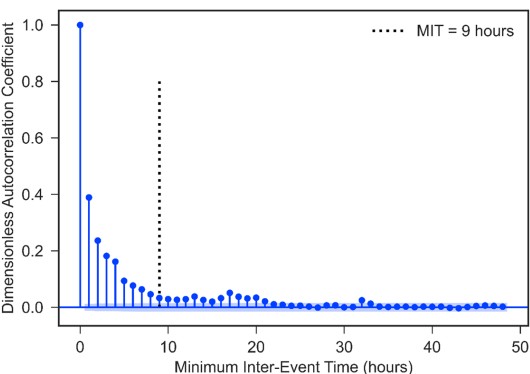

**Figure 8.** Autocorrelation of rain pulses

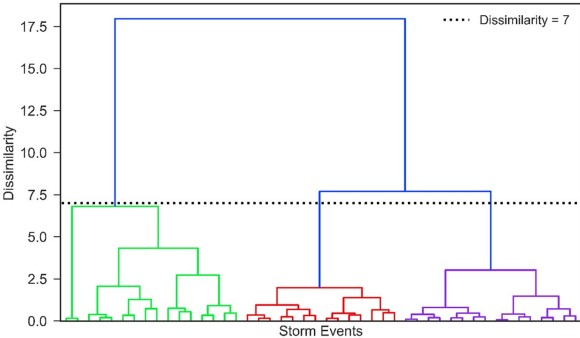

**Figure 9.** Tree-based dendrogram of agglomerative clustering

accumulated precipitation; average hourly intensity; and maximum 15-minute, 30-minute, and one-hour interval intensities
were computed. These features were then tested for their utility in generating independent storm clusters with comparable
storms. The maximum 15-minute interval intensity and the total accumulated precipitation were found to generate the most
comparable storms with agglomerative clustering. To define the optimum number of clusters, Ward linkage method was used
to minimize the total within-cluster variance (Edelbrock, 1979). Figure 9 shows a dendrogram that illustrates how clustering
the storms into three groups captures acceptable dissimilarity between storms, which are defined as light, moderate, and severe
storms. The vertical axis of the dendrogram depicts the dissimilarity between storms, and the horizontal axis represents storms.
The position of each split on the vertical axis shows the dissimilarity of the two clusters on sides of the split. Table 3 shows
summary statistics for the three storm clusters.

**4.3   Waze Data Preprocessing**

Potential flood-related alerts posted in the timespan of 2018-04-21 to 2020-03-20 were matched to their most recent preceding
storm, and the delay between the time of each alert's posting and the end of rainfall was calculated. Figure 10 gives the





**Table 3.** summary statistics of storm clusters

| Storm cluster | Number of storms | Mean of Maximum 15-min intensity (in/15 min) | Mean of total precipitation (in) | Mean of duration (hours) |
|---|---|---|---|---|
| Light | 142 | 0.05 | 0.12 | 4.08 |
| Moderate | 70 | 0.29 | 0.85 | 8.89 |
| Severe | 24 | 0.72 | 2.99 | 18.59 |

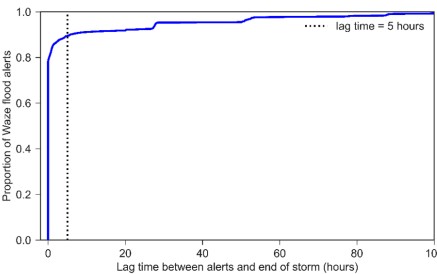

**Figure 10.** Distribution of delay in alert posting from storm end

distribution of delays between alert's published time and storm end. Figure 10 shows that more than %90 of Waze flood alerts are posted within 5 hours of storms. Therefore, potential flood-related alerts posted later than 5 hours after storms were
considered outliers (noise) and removed from the analysis. This processing left 4,996 flood-related alerts out of the initial 5,652 alerts. The number of flood alerts posted per storm event ranged from 0 to 375, with the distribution depicted in Figure 11. During the study period, 150 storms occurred but only 98 storms caused Waze flood alerts. On average, each storm event had ten flood alerts. Among the 4,996 flood alerts that were filtered, 2,665 alerts were assigned to 191 independent surface depressions using the approach described in the methodology section.

**5   Results**

The performance of the proposed framework in estimating flood frequency is evaluated using both the Empirical Bayes (EB) and Random Forest (RF) models and compared to the baseline models. Results from the best-performing model, EB, are then examined in more detail in the following sections.

**5.1   Model Parameters and Performance**

A random %80-%20 train/test split is implemented to evaluate models. Models are fit using a randomly-selected training dataset that represents %80 of the processed flood alert dataset, with the remaining %20 of the dataset (the testing dataset) used for assessing the predictive power of the models. Parameters for the fitted NB model (Equation 4) are presented in Table 4.

The dispersion parameter of the fitted NB regression model ($\phi$ of Equation 3) is 2.943. A value of $\phi > 1$ demonstrates that the over dispersion assumption is valid, whereas $\phi < 1$ shows an under-dispersed dataset. The MAE value achieved from fitting





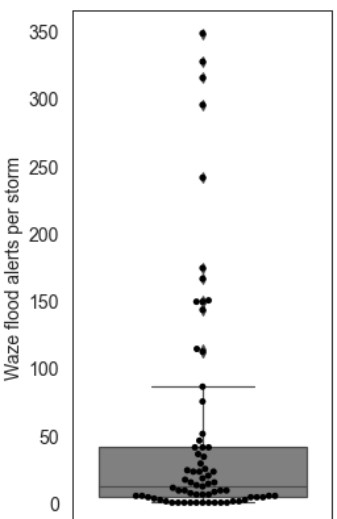

**Figure 11.** Total number of flood-related alerts per storm

**Table 4.** NB model estimation results

| Variable | Coefficient | Standard Error | Z value | P-value |
|---|---|---|---|---|
| Constant | -9.30E+01 | 3.17E-01 | -295.3 | 0.000*** |
| Moderate storm | 5.10E+01 | 2.10E-02 | 3.1 | 0.000*** |
| Severe storm | 7.60E+01 | 2.40E-04 | 3.3 | 0.000*** |
| Net DA | 8.10E-03 | 1.20E-01 | 445.9 | 0.001** |
| Average slope | 6.30E-02 | 1.10E-01 | 686.2 | 0.003** |
| Log ADT | 8.20E-02 | 2.70E-02 | 3 | 0.003** |
| Goodness of fit | | | | |
| BIC | | 1836.31 | | |
| MAE | | 1.74 | | |
| *** significant with a 99% confidence interval | | | | |
| ** significant with a 95% confidence interval | | | | |

the NB distribution is 1.74, which shows that the flood frequencies fit to the prior probability distribution have an average error equal to 1.74 flood events out of 150 storms. The EB estimate of the fitted NB regression model, computed based on Equation 6, reduces the MAE on the training set to 0.88 flood events.

For the RF model, hyperparameter tuning is implemented using a 3-fold cross-validated randomized search in the Scikit-Learn library in Python programming environment. The best-performing model is found to have ten trees. The features with the highest importance (based on impurity-based feature importance calculated by the Scikit-Learn library) in the RF model





are severe storms, maximum depth, average upstream slope, logADT, and the net drainage area. The MAE of RF estimates on the training set is 0.73.

The predictive power of both models is evaluated on the held-out test dataset. The EB approach predicts the number of flood events for unseen situations with MAE=0.92, while the RF model's evaluation MAE is considerably higher, with MAE=2.1.
To minimize the impact of particular train-test datasets on the model's performance, the dataset is randomly split 50 times and the model performance statistics are re-evaluated for each split. Table 5 compares statistics on EB and RF model performances for 50 runs. Figure 12 shows the prediction power of the models on the train and test datasets.

**Table 5.** Average predictive power on random test sets

|  | Empirical Bayes | | Random Forest | |
| --- | --- | --- | --- | --- |
|  | Average | Standard deviation | Average | Standard deviation |
| MAE | 0.89 | 0.11 | 1.92 | 0.18 |

It can be seen that the RF model is a better fit on the training dataset but its lower performance on the test set shows that it is overfitting on the training set while the EB approach has more consistent performance on both datasets. The superiority of
the EB model shows that the unobserved features play a significant role in PFF formation on road segments and a Bayesian approach is more successful in capturing the effects of these features.

Next, the EB model that is found superior to the RF model is compared with the simple benchmark models given in the methodology section. Figure 13 demonstrates how the flood counts will be predicted on the test dataset using each benchmark model, NB regression, and EB model. Table 6 summarizes the performance of the EB approach, NB regression, and benchmark
models. It can be seen that the MAE for both training and testing sets improves by adding storm clusters to the average model. This increase is more noticeable in light storms (almost %50 improvement for both training and testing dataset).

However, adding topographic and observed flooding variables, as in the EB model, increases the accuracy of PFF count estimation for severe storms more than moderate and light storms. This shows that topographic features are more important in the formation of PFF when storms are more severe. Also, if PFF is observed at a particular location, then it is more likely to be observed at that depression again.

**Table 6.** Physical depression/catchment descriptors

|  | MAE of train set | | | | MAE of test set | | | |
| --- | --- | --- | --- | --- | --- | --- | --- | --- |
|  | Light | Moderate | Severe | Total | Light | Moderate | Severe | Total |
| Total average | 1.88 | 2.01 | 2.53 | 2.14 | 2.19 | 1.93 | 3.04 | 2.37 |
| Storm cluster based average | 0.95 | 1.97 | 2.52 | 1.82 | 1.16 | 1.86 | 2.72 | 1.89 |
| NB regression | 0.94 | 1.91 | 2.37 | 1.74 | 1.16 | 1.65 | 2.75 | 1.82 |
| Empirical Bayes | 0.69 | 1.01 | 0.93 | 0.88 | 0.84 | 0.85 | 1.09 | 0.92 |



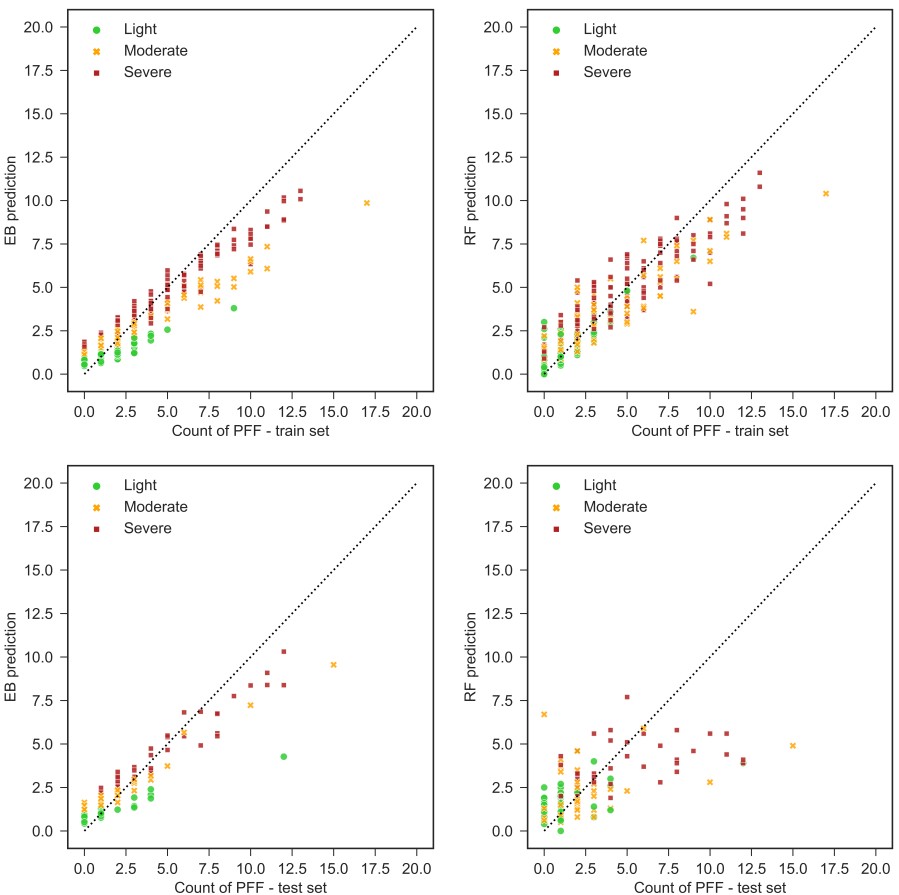

**Figure 12.** Prediction of number of roadway PFF events

## 5.2 Flood Likelihood Estimation

The EB approach is superior in predicting the total number of flood events; hence, this approach is used to estimate flood likelihoods from the frequency of PFF events (Equation 11). Figure 14 shows the distribution of flood likelihoods for each storm type across all depressions. As expected, we see a higher PFF likelihood during severe storms compared to light and moderate storms. Figure 15 shows how likelihoods match with flood events that were posted to the Waze dataset. Generally, we can see that flood likelihoods are higher when flooding has been posted. However, as discussed in the methodology section, true negative situations cannot be identified with voluntary crowdsourced data (i.e. there could be flooding that no Waze user has reported).

Flood maps that predict the probability of flooding for each depression are then derived for each storm cluster. Figure 16 shows an example of a flood probability map for severe storms, along with historical flood-related alerts and traffic jams reported by Waze during one particular severe storm occurred on October 8th, 2018. Waze traffic jam reports include severity

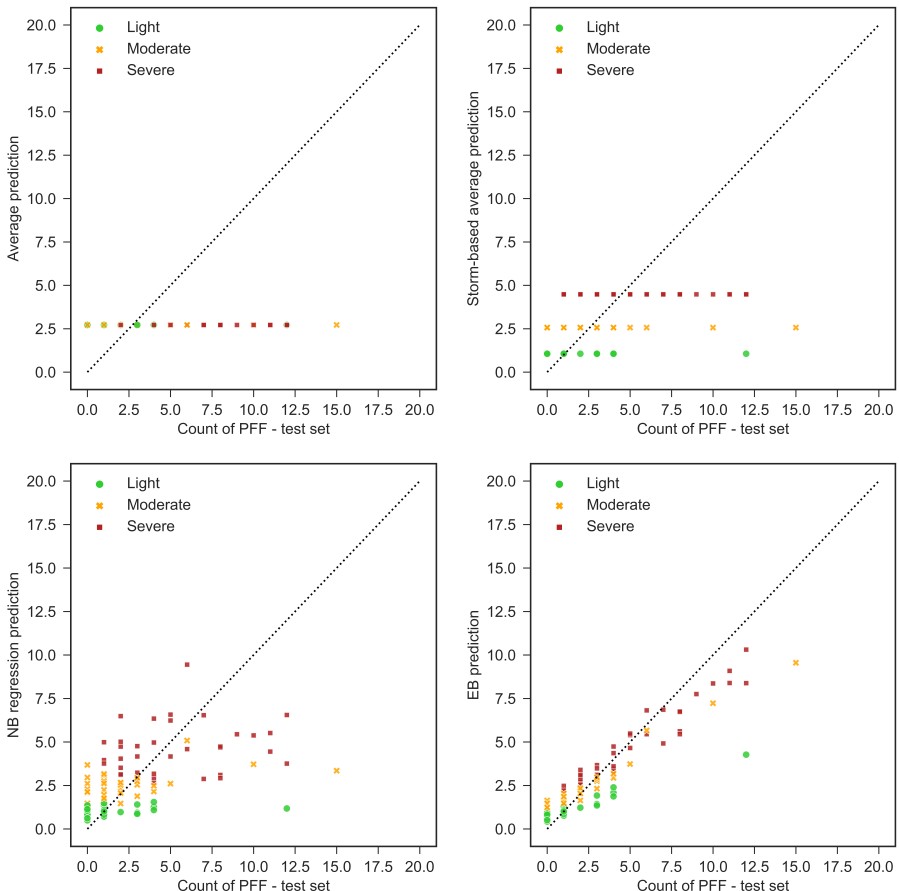

**Figure 13.** Prediction of number of PFF using benchmark models and EB

and congestion levels ranging from 1 (lowest) to 5 (highest), which denote the level of traffic slow down or complete shutdown. Negligible, low, moderate, and high flood probabilities are defined as less than %10, less than %30, less than %50, and higher than %50, respectively. In Figure 16, high traffic levels (Waze jam levels of 5) can be seen near a depression with high PFF
probability (more than %50).

## 6   Discussion

The EB model is superior compared to the RF and benchmark models in predicting the number of flood events; hence this model is used to estimate flood probabilities for storm clusters. The distribution of estimated flood probabilities (Figure 14 and Table 6) are plausible given the magnitude of the storms. For example, the light storms have average duration of 4 hours
and average total precipitation of 0.1 inches, which is quite low and flooding would not be expected during these storms. Flood-related alerts that are posted during these rainfall events can be assumed to be noise and disregarded for future studies.


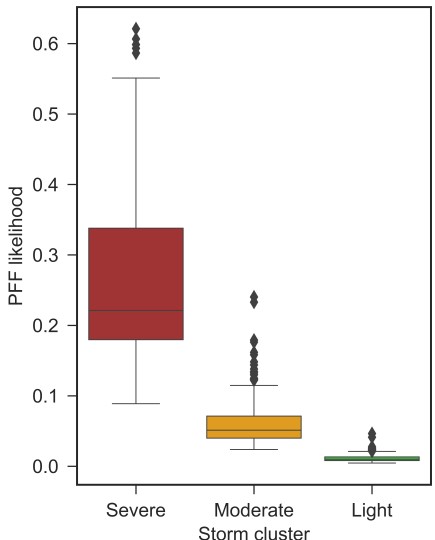

**Figure 14.** Distribution of flood likelihoods for light, moderate, and severe storms at depressions in the study area

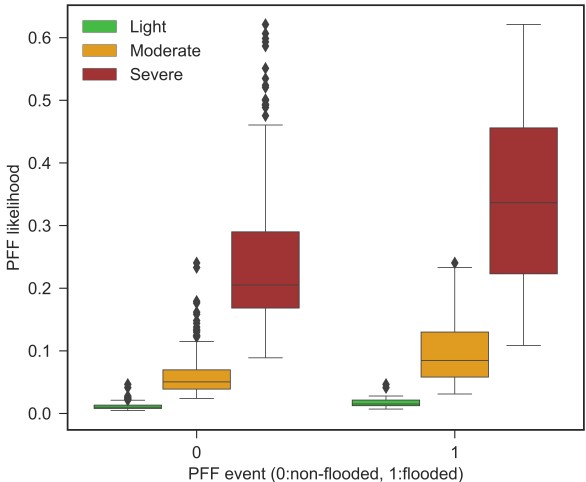

**Figure 15.** Distribution of flood likelihoods in reported versus non-reported floods

Based on the NB regression line that is fitted to the count of observed flood events, we expect to see 7.6 and 5.2 times more flood events in moderate and severe storms, respectively, compared to light storms. The NB model also shows that increases in the upstream net drainage area and average slope increase the probability of flooding, as would be expected. Furthermore, log ADT has a direct relationship with the probability of observing a PFF event because frequently-traveled roads are more likely to have Waze postings. This finding shows the limitations of estimating flood events from crowdsourced Waze datasets that




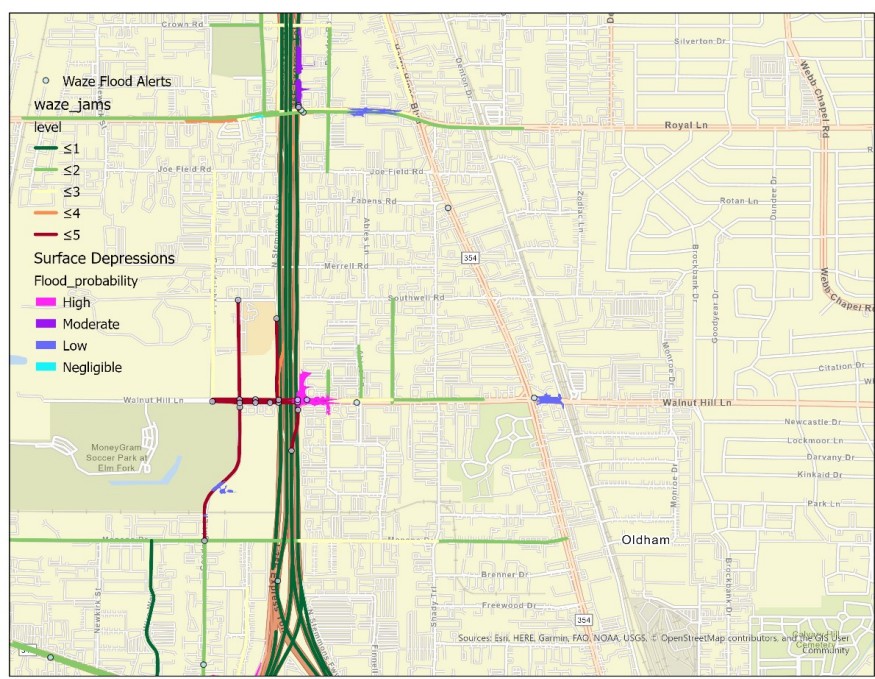

**Figure 16.** Severe storm PFF probability map versus flood alerts and traffic jams on October 8th, 2018

tend to neglect flood events on less-traveled roads. The superior performance of the EB approach shows the significant impact of unobserved site-specific features such as stormwater inlet conditions in predicting the likelihood of PFFs on roadways. By using historical observations, the EB approach better identified frequently-flooded locations (road surface depressions), perhaps

due to site-specific features such as under-sized stormwater inlets. Data were not available on these features for this study. In highly urbanized areas, these types of uncertainties in engineered structures, particularly in older areas of the city where recordkeeping can be poor, add to temporal uncertainties such as changing climate and land use that can affect flood formation. Despite these limitations, this study showed that localized traffic-related flood alerts are helpful in estimating PFF probabilities over a three-year period. For longer periods, periodically retraining the model to account for changes in infrastructure and

climate is recommended.

To make effective use of crowdsourced traffic data, extensive preprocessing is needed to evaluate the reliability of the data and map flood alerts, which are not necessarily posted at the exact location of the flooding, to plausible nearby depressions. This processing, which was done manually in this study, can introduce errors and bias to the analysis. With more data and integration of other data sources (e.g., flood sensors), an automated mapping process could be developed that would likely

reduce these errors.

Furthermore, the approach taken in this study only considers flood-prone locations reported by Waze users. Numerous parameters affect human exposure to flooded locations, such as the number of Waze users that pass a road segment, road





type, road function, day of week, and time of day. Hence, a similar flood extent on the road can cause significantly different magnitudes of traffic disruption at different times and locations, and, therefore, different flood reports. To develop a more
unbiased flood prediction model, we suggest that crowdsourced data be used as complementary data in conjunction with other data sources and models to account for less frequently traveled areas and times (e.g., during the Covid-19 pandemic, which was not included in this study when traffic was significantly reduced).

## 7  Conclusion

This analysis is a first step in exploring approaches to implement crowdsourced data from the Waze app into flash-flood
prediction. For this case study, Waze flood alerts were primarily posted in areas outside of mapped river flood hazards and low water crossings, suggesting the need for and importance of modeling rainfall-induced or pluvial flash flooding (PFF). The statistical and ML models implemented in this study demonstrated the feasibility of modeling PFF in terrain depressions based on storm, catchment, and road properties. The EB approach is found to be superior in terms of predictive power compared to RF, which shows the importance of unobserved site-specific features on roadway PFF, which the EB approach can better
consider using a Bayesian approach to historical flood events. Both statistical and machine learning models achieve smaller MAEs for severe storms compared with moderate and light storms. This shows that the modeled depression and catchment descriptors are more explanatory in severe storms when infiltration is reduced and drainage systems are more likely to be overwhelmed. The high accuracy of the proposed methodology in the Dallas case study shows that crowdsourced traffic data has value for high spatio-temporal resolution flash flood prediction, but further research is needed to more fully exploit its use
as a complementary data source with more official data sources and physics-based models.

*Acknowledgements.*  This research was funded by the U.S. Department of Commerce, National Institute of Standards and Technology (NIST) Public Safety Innovation Accelerator Program (PSIAP) under award number 60NANB17D180. We gratefully acknowledge NCTCOG for granting us access to their Waze data. We also acknowledge the Dallas Fire Rescue Department for their collaboration in defining and reviewing this research.





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
