# Peer review of "Estimating the likelihood of roadway pluvial flood based on crowdsourced traffic data and depression-based DEM analysis"

_Natural Hazards and Earth System Sciences, 2022_

## Author Response (AR1)

**Responses to RC1:**

**Enclosure:**

Response letter to the reviewers' comments

-  texts are removed from the manuscript
- Green texts are added to the manuscript

1. **In Section 2.1.1 the authors describe the process for extracting depressions. This is an important and challenging step given the heterogeneity of the urban environment and noise present in high resolution DEM's. The authors manually identify the smallest meaningful flood prone depression. Can you elaborate on the heuristics used to make this choice? For example, if a researcher were to repeat this process in another city, how would you guide them? Also is there any potential to automate this step in the future by perhaps brining in other spatial information such as shape files of roadways?**

Response:

We identified depressions, and their associated depression levels, that were close to or larger than the scale of a road and could affect traffic flow. At present this is a manual process, though it may be possible to develop an automated process that selects depression levels by eliminating levels that result in depressions smaller than the size of a road or traffic lane. We've revised the manuscript at line 122 to clarify this:

"Due to the complexity of urban terrain, the spatial scale of depressions at each hierarchy level is quite variable, and depressions at the same level can be as large as a neighborhood or as small as a pothole.  Initially, depressions at all hierarchical levels are extracted, and the level that has depressions that are at the scale of, and best align with urban features, including roadway curbs and gutters, was manually selected. Flood-prone depressions are then identified by examining overlays of the depressions and Waze flood reports, as well as the areas of depressions and road surfaces that the depression covers. The procedure for using Waze reports to identify depressions is presented in detail in Section 2.1.5."

2. **On line 193 the authors mention that several individuals made assessments of which flood alerts to assign to which depressions. Please elaborate on this process. Did these several individuals making the determination together? Or did these individuals make their assessment separately? If it was the later, how much agreement was there between assessments and how did the research team make the final determination? Was this process followed for all 4,996 Waze alerts in the Dallas case?**

Response:

The methodology section is updated to emphasize that individuals have performed their analysis separately.

Line 193:

"In this study, several independent individuals were asked to visually assess a map of historical flood alerts laid over surface depressions and assign alerts to depressions separately using the following criteria: a cluster of more than two flood alerts should be available near the depression and the depression must be distinct from other nearby surface depressions. Flood alerts posted from bridges and elevated highways are excluded since BE-DEM does not represent bridge surfaces."

To present the level of agreement between researchers alert to depression assignment, the following text is added to the manuscript in the results section:

Line 388:

The process of flood alert assignment explained in the methodology section (Section 2.1.5) was performed for the 4,996 flood alerts in the Dallas case study by several independent individuals. With the criteria given previously, there was 90.5% agreement between the annotators in the assignment of alerts to depressions. The first author reviewed alerts that indicated disagreement and if the specified criteria for making the assignment were met, she completed the assignments using best judgment. Among the 4,996 flood alerts that were filtered, 2,665 alerts were assigned to 191 independent surface depressions using the approach described in the methodology section (Section 2.1.5).

3. **In the EB model, how was the weighting factor, w, determined? Is w also a calibrated parameter?**

Response:

"$w$" is a function of the negative binomial distribution parameters (Equation 3). The manuscript is updated as follows to make it more transparent.

$$Equation\ 2: \quad EB(y) = w \times \mu + (1 - w) \times y$$

$$\text{Equation 3:} \quad P(y) = \frac{\Gamma(y + \phi)}{\Gamma(y + 1)\Gamma(\phi)} \left(\frac{\phi}{\phi + \mu}\right)^{\phi} \left(\frac{\mu}{\mu + \phi}\right)^{y}$$

Line 255:

It can be shown that the weight $w$ in the EB equation based on the NB regression is calculated as $\phi/\mu + \phi$, hence we can rewrite Equation 2 as Equation 6. $\phi$ is the NB parameter (Equation 3) estimated using Maximum likelihood estimation. For more information regarding the mathematics of deriving the EB weight factor, refer to Zou et al. (2017).

$$\text{Equation 6:} \quad EB(y) = \frac{\phi}{\mu + \phi}\mu + \frac{\mu}{\mu + \phi} y$$

4. **In section 2.1.4 of the methodology it was not clear how were three precipitation categories selected. The authors mention agglomerative clustering but not what selection criteria was used. The criteria is mentioned in the Case study pre-processing section, but I recommend moving it up to the methods section.**

Response:

To address this comment, the following text is replaced from line 471 to line 173 of the manuscript:

To define the optimum clusters, the Ward linkage method was used to minimize the total within-cluster variance (Edelbrock, 1979). In this method, increases in within-cluster variance are minimized to find the optimum pair of clusters to merge.

5. **In the discussion section, please include a discussion of the limitations of the data sets and models presented.**

Response:

The discussion section is updated to reflect the limitations of the dataset and models as follows:

Added to line 469:

Furthermore, the approach taken in this study only considers flood-prone locations reported by Waze users. Numerous parameters affect human exposure to flooded locations, such as the number of Waze users that pass a road segment, road type, road function, day of week, and time of day. Hence, a similar flood extent on the road can cause significantly different magnitudes of traffic disruption at different times and locations, and, therefore, different flood

reports. Data-driven models also have limitations due to the previously discussed dataset constraints.

The EB model accounts for heterogeneity by utilizing historical frequencies. However, this does introduce a bias towards more frequently traveled routes, as discussed in Section 2.1.5, and the EB model estimates will be skewed and less accurate for depressions situated on local and less-traveled routes. While major routes are more important than minor routes for minimizing exposure and risk, we do acknowledge this as an unavoidable limitation. It is possible that, with more data, an approach to extrapolating findings on major roads to minor roads could be developed. To develop a more unbiased flood prediction model, we suggest that crowdsourced data be used as complementary data in conjunction with other data sources and models to account for less frequently traveled areas and times (e.g., during the Covid-19 pandemic, which was not included in this study when traffic was significantly reduced).

Minor comments:

1. **Line 40, in addition to speed limits when driving through water, full loss of control is also possible. "As little as one foot of water can move most cars off the road." NWS 2011.**

Response:

The introduction is updated as follows:

Line 42:

For example, Pregnolato et al. (2017) estimated that a driver facing 10 cm of standing water must not drive faster than 40 km/hr to maintain safe driving, stopping, and steering without loss of control. Furthermore, according to the National Weather Service (NWS 2011) 30 cm of standing water can be sufficient to float most cars.

2. **In preprocessing the Waze data (Section 2.1.5), is there information on direction of travel? If so, is that information used to constrain the possible flooded locations?**

Waze data do not provide the direction of travel. However, no constraints regarding the travel direction have been used for assigning flood alerts to flooded depressions, since depressions can cross both sides of the road. The methodology section is updated as follows to clarify that travelers might post a flood alert on either side of a flooded location.

Line 191:

Posting a flood alert requires Waze users to complete three steps (three selections) in the app while driving or riding, and, as a result flood alerts may be posted some distance along the

roadway in either direction from the flooded road segment. Waze data do not provide direction of travel; Hence assigning flood alerts to the proper depression must be done carefully."

3. **In Figure 1, why does the last bullet point of the central section read "Alerts/depressions." Please clarify.**

It is changed to "Assignment of alerts to depressions":

[Figure]

*Figure 1. Methodology*

4. **Figure 6: It is hard for the viewer to make accurate comparisons between pie charts (see Helsel et al. 2020). I suggest replacing this figure with a bar graph. Additionally, the font sizes vary notable between Fig 6a and 6b.**

The figure is changed to a bar chart, shown below.

[Figure]

*Figure 6. a: Distribution of NFHL flood zone areas across the study region, b: Flood alerts in NFHL flood zones*

5. **A sentence in the text could substitute for Table 5.**

Table 5 is removed and the following changes are made in Lines 411:

To minimize the impact of particular train-test datasets on the model's performance, the dataset is randomly split 50 times and the model performance statistics are re-evaluated for each split.  The EB model has an average MAE of 0.89, as opposed to the average MAE of 1.92 attained by the RF model. EB's predictive capability is also more stable across the 50 runs than the RF model, with MAE standard deviations of 0.11 and 0.18, respectively.

6. **Add the results for the RF model to Table 6 as well for comparison.**

The table is changed and the updated version is shown below.

| | MAE of train set | | | | MAE of test set | | | |
|---|---|---|---|---|---|---|---|---|
| | Light | Moderate | Severe | Total | Light | Moderate | Severe | Total |
| Total average | 1.88 | 2.01 | 2.53 | 2.14 | 2.19 | 1.93 | 3.04 | 2.37 |
| Storm cluster based average | 0.95 | 1.97 | 2.52 | 1.82 | 1.16 | 1.86 | 2.72 | 1.89 |
| NB regression | 0.94 | 1.91 | 2.37 | 1.74 | 1.16 | 1.65 | 2.75 | 1.82 |
| Empirical Bayes | 0.69 | 1.01 | 0.93 | 0.88 | 0.84 | 0.85 | 1.09 | 0.92 |
| Random Forest | 0.68 | 0.98 | 0.91 | 0.86 | 1.34 | 1.66 | 2.76 | 1.92 |

7. **Figure 16: include numeric probabilities associated with high, moderate, etc. flooding on the figure or in the caption.**

The figure is changed, with the new version shown below.

[Figure]

*Figure 16. Severe storm PFF probability map versus flood alerts and traffic jams on a. Friday, September 22nd (the date of a severe storm), b. Friday, September 29th, 2018. and c. Friday, September 15th, 2018*

**Responses to RC2**

**Enclosure:**

Response letter to the reviewers' comments

-  texts are removed from the manuscript
- Green texts are added to the manuscript

1. **The manuscript is difficult to follow. There are pieces of information on the same thing throughout the manuscript rather than organized in the same section. As a result, I had to read back and forth to get a clear view. For example, depression extraction is described in Sections 2.1.1 and 4.1 storm event clustering is described in Sections 2.1.4 and 4.2. It would be easier for the reader to follow the methodology if the information were presented concisely and organized.**

To address these concerns, we've revised the manuscript to put all of the method details in section 2 and leave section 4 for results. Specifically, the methodology is explained in Sections 2.1.1 and 2.1.4, while the results are presented in Sections 4.1 and 4.2. Sentences that repeat methodology are removed from Sections 4.1 and 4.2. Lines 367, 372, 375, and 378 are updated.

2. **Section 2.1.1: It is vague how depth extraction was used to find flooding that can cause travel disruption. Was the depth of depressions at L2-1 and L2-2 considered to determine if there was any impact? The depth of flooding at the different levels of depressions is not mentioned in the manuscript.**

We included only the depressions, as well as the depression levels, that were close to or larger than the scale of a road and could affect traffic flow. Depth of depressions is not used to find flooding. Depressions that could cause flooding are identified by visually investigating depressions' alignments on the road surface, micro topographic features and their area, as well as whether a cluster of Waze alerts is assigned to them. In the example presented in Figure 2, L2-2 is not chosen as roadway flooding because it does not overlay the road surface. Since the depth and level threshold criteria have not been explained in the manuscript before line 122, the sentence shown in red below is removed to resolve the confusion.

"Due to the complexity of urban terrain, the spatial scale of depressions at each hierarchy level is quite variable, and depressions at the same level can be as large as a neighborhood or as small as a pothole.  Initially, depressions at all hierarchical levels are extracted, and the level that has depressions that are at the scale of, and best align with urban features, including roadway curbs and gutters, was manually selected. Flood-prone depressions are then identified by examining overlays of the depressions and

Waze flood reports, as well as the depression area and road surfaces that the depression covers. The procedure for using Waze reports to identify depressions is presented in detail in Section 2.1.5."

3. **Crowdsourced flood reports like Waze often have multiple reports surrounding a flooded area for the same event. There was no mention of removing duplicates in the manuscript. Not removing duplicates could falsely increase the probability of flooding on a depression.**

Reporting of depression flooding is considered as a binary presence/absence variable and encoded as 1 if there is one or more flood alert and 0 if there is no flood alert. This process removes duplicates. To clarify this, the following statement is added to the manuscript.

Line 201:

Pluvial flooding on any given surface depression can be modeled as a Bernoulli trial of flood failure (i.e., non-flooded) or success (i.e., flooded). If a depression has one or more Waze flood alerts linked to it, the depression is labeled as flooded (success). Assuming that the probability of being flooded is smaller than the non-flooded situation and that the likelihood of flooding in a particular storm event for each depression is independent of the probability of flooding for other depressions, a random variable $y_{i,j}$ will define the count of successes (flooding) out of the N trials (N storm events of cluster j) on depression $i$.

4. **Line 427-431: The comparison is not clear. Equation 11 used "yi,j is the predicted number of floodings on depression i and storm type of j". Isn't the predicted number of flooding derived using historical flood reports? In that case, the likelihood of flooding should be higher when there were Waze reports. If the purpose is to evaluate model performance in predicting flood probability, some performance metric should be used.**

As the reviewer notes, the predicted number of floodings is derived from historical flood reports, which means that the likelihood of flooding should be higher when there were Waze alerts posted. Figure 15 is intended only to demonstrate that the predicted likelihoods are plausible, but we cannot evaluate model performance explicitly since the data do not contain true negatives (i.e., flooding may be occurring in locations where there are no Waze reports).

Note that in responding to this comment, the authors realized that the description of Figure 14 could easily be made just with Figure 15. Therefore Figure 14 is removed from the revised manuscript.

5. **Line 437-440 and Figure 16: This part needs further clarification. The probability of flooding and jam could be shown of two maps, if they are overlapped. Jam level 3 is not clear on the map. Please consider changing the color. The jam during the storm**

**event should be compared with the jam during the same time and same days to conclude it happed due to the flooding.**

The map layout and colors are changed to make the levels clearer, along with zooming in to the example intersection. Visual comparison between the traffic jams at the same time and day of the week for the following and preceding weeks are added to the map. The manuscript is updated as follows:

[Figure]

*Figure 16. Severe storm PFF probability map versus flood alerts and traffic jams on a. Friday, September 22nd (the date of a severe storm), b. Friday, September 29th, 2018. and c. Friday, September 15th, 2018*

Line 434:

Figure 16-a shows an example of a flood probability map for severe storms, along with historical flood-related alerts and traffic jams reported by Waze during one particular severe storm that occurred on September 22nd, 2018. Figures 16-b and 16-c show the same information during the same time and day of the week for the following and preceding weeks. Waze traffic jam reports include severity and congestion levels ranging from 1 (lowest) to 5 (highest), which denote the level of traffic slow down or complete shutdown. Negligible, low, moderate, and high flood probabilities are defined as less than %10, less than %30, less than %50, and higher than %50, respectively. In Figure 16-a, high traffic levels (Waze jam levels of 5) can be seen near a depression with high PFF probability (more than %50). Figure 16 indicates that traffic jams during severe storm are noticeably higher than at similar time intervals before and after the storm. These maps suggest that the traffic jam on the storm date, which agrees with the flood likelihood, is likely to be an anomaly relative to typical traffic conditions at this intersection. This finding is consistent with the flood alerts and predictions of severe flooding at this location during the storm.

Minor Comments:

1. **Line 22-23: This sentence should go to the last paragraph of Introduction. The first paragraph in Introduction usually provides a general background of the problem rather than specifying the goal of the study at the very first sentence.**

The authors believe that providing the goal of the study at the start of the paper helps to focus the reader on the relevant portions of the background and motivation. We would prefer to leave the sentence where it is, but if the editor prefers that we make this change then we will do so.

2. **Table 2. There is no definition of the storm clusters until Section 4, line 376.**

To address, line 170 is updated as follows:

Depending on storm events' severity and terrain characteristics, storms can produce similar patterns of depression PFF. To capture this phenomenon, storms are clustered into classes based on their severity (light, moderate and severe) using storm characteristics. For storm clustering, agglomerative hierarchical clustering is applied using a bottom-up approach that forms a single cluster for each storm event and successively merges clusters with the smallest distances between features. The benefit of using agglomerative clustering is that this algorithm is less sensitive to outliers (Edelbrock, 1979).

---

## Author Response (AR2)

I thank referees and editor for reviewing our manuscript and providing valuable comments and suggestion.

Enclosure:
The track changed manuscript is provided. Yellow highlights are text added to the manuscript and Red highlights are removed from the manuscript.

Responses to RC1:

- On lines 125-126 inches should be converted to metric to match the rest of the paper
    o Thanks for pointing it out. Feet is changed to centimeters.

- On line 203 the word separately is missing (the response to reviewers indicated it was added it is not here)
    o Manuscript is updated with the word separately added.

- On line 270, only need to define variables once
    o Repetitive variable definitions are removed. Thanks for noting.

- On line 330, $y_{i,j}$ is missing the hat
    o Manuscript is fixed

- Section 4 presents results. I recommend integrating this into the results section
    o Section titles are updated. Since the framework has two main parts of preprocessing and modeling, authors decided to keep them in two separate sections to make it easier to understand. As reviewer suggested, section titles are changed to preprocessing results and modeling results.

- In the caption for Figure 9 note what the colors refer to.
    o Caption is updated in the revised manuscript.

- The first two sentences of section 5.1 are methods not results and could be removed.
    o The method descriptions are removed.

- There is a comma missing after important on line 488
    o The manuscript is updated.

RESTRICTED